# Research of Frequency Splitting Caused by Uneven Mass of Micro-Hemispherical Resonator Gyro

**DOI:** 10.3390/mi13112015

**Published:** 2022-11-18

**Authors:** Lijun Song, Qingru Li, Wanliang Zhao, Tianxiang Zhang, Xing He

**Affiliations:** 1School of Information and Control Engineering, Xi’an University of Architecture and Technology, Xi’an 710055, China; 2Shanghai Aerospace Control Technology Institute, Shanghai 201109, China

**Keywords:** micro-Hemispherical Resonator Gyro (m-HRG), frequency splitting, uneven mass, the partial differential equation, high-quality factor

## Abstract

In practical engineering, the frequency splitting of Hemispherical Resonator Gyro (HRG) caused by uneven mass distribution seriously affects the precision of HRG. So, the inherent frequency is an important parameter of micro-Hemispherical Resonator Gyro (m-HRG). In the processing of hemispherical resonator, there are some morphological errors and internal defects in the hemispherical resonator, which affect the inherent frequency and the working mode of m-HRG, and reduce the precision and performance of m-HRG. In order to improve the precision and performance of m-HRG, the partial differential equation of the hemispherical resonator is solved, and the three-dimensional model using ANSYS software accurately reflected the actual shape is established in this paper. Then, the mode of hemispherical resonator in ideal state and uneven mass distribution state are simulated and analyzed. The frequency splitting mechanism of the hemispherical resonator is determined by calculation and demonstration, and the frequency splitting of the hemispherical resonator is suppressed by partial mass elimination. The results show that the absolute balance of energy can ensure the high-quality factor and the minimum frequency splitting of the hemispherical resonator. Therefore, during the processing of hemispherical resonator, the balance of mass should be achieved as much as possible to avoid various surface damage, internal defects and uneven mass distribution to guarantee the high-quality factor Q and minimum frequency splitting of hemispherical resonator.

## 1. Introduction

HRG is a new type of solid wave gyro with high precision and reliability, which has developed rapidly in recent years. Compared with ring laser gyro (RLG) and resonator micro-optical gyro (RMOG), it has the advantage of wide angular rate output range and can meet the requirements of precise positioning, high-speed cruise and precise strike of flying missiles in the national defense field. Compared with optical gyro, HRG has simple structure, is not easy to damage, and is more durable than other high precision gyros, its key component is high-precision hemispherical resonator made of high-quality fused quartz glass, References [1,2] also analyze the frequency stability, precision spectroscopy and quantum applications of the hemispherical resonator in detail. The manufacturing precision of HRG is restricted by the manufacturing accuracy of hemispherical resonator. In particular, the removal, migration or addition of materials at the atomic scale is the key factor to realize the absolute rotational symmetry of hemispheric resonator.

Ren, S.Q. et al. analyzed the frequency splitting caused by the uneven density of hemispheric resonator, but the deriving of natural frequency splitting is too complicated and abstract [3]. Chen, X. et al. analyzed the angular rate error of hemispheric resonator caused by uneven thickness of the thin wall, but does not analyze the influence of uneven thickness quality on its performance. According to the principle of error minimizing, the dynamic equation of the second-order natural mode of hemispherical resonator with uneven density distribution is established. The mechanism of frequency splitting by the fourth harmonic of density is analyzed by deriving the partial differential equation of uneven density circumferential distribution, the modes of uneven mass are simulated and analyzed, which provides new ideas to reduce the zero drift of HRG [4].

## 2. The Frequency Splitting Caused by Uneven Mass

The uneven mass of the hemispherical resonator will lead to the frequency splitting of the two modes of HRG, and the uneven mass distribution is mainly caused by the manufacturing process defect of the hemispherical resonator. Therefore, the high-precision machining of the hemispherical resonator is the most critical and difficult process in the manufacturing process of the hemispherical resonator gyroscope [5,6]. In the processing of the hemispherical resonator, the mass should try to achieve balance, and avoid various surface damage, internal defects and uneven mass distribution, so as to prevent affecting the precision and performance of hemispherical resonator gyroscope.

### 2.1. The Theory of Frequency Splitting

Frequency splitting is the difference between the resonant frequencies of two degenerate modes when the hemispherical resonant gyro operates in two four antinode vibration modes. The ideal hemispherical resonator is a perfectly symmetrical structure of eight centers and one line, the driving mode and the sensitive mode are perfectly matched in the same order and orthogonal state; therefore, the problem of frequency splitting is averted [7].

The hemispherical resonator is the core device that affects the measurement precision of m-HRG, especially its structural design, machining and other key factors, which is the technical bottleneck that restricts the development and practical engineering application of m-HRG. In the high-precision machining of hemispherical resonator, due to the limitation of process level and precision of machining equipment, there is an uneven circumferential distribution of the mass and damping of the hemispherical bare resonator, which leads to frequency splitting of m-HRG.As shown in Figure 1, the two modes have two natural axes 45° apart from each other. The axis with higher natural frequency is the large stiffness axis. The axis with lower natural frequency is the small stiffness axis.

### 2.2. The Partial Differential Equation of Hemispherical Resonator with Ring Structure

The ring model of hemispherical resonator is to divide the hemispherical resonator into multiple sections along the direction of the support rod. In each cross section, the geometric structure of the hemispherical resonator can be approximately regarded as a thin elastic ring. Hemispherical resonator is a thin-walled hemispherical shell fixed on the support rod, which is made of fused quartz glass with high precision. Its thickness error is less than 2 μm, the flatness error is less than 2 μm, and the surface roughness Ra value is better than 2 nm. In order to simplify the dynamic analysis of hemispherical resonator, the vibration of m-HRG can be simplified as the vibration of thin elastic ring. The schematic diagram of thin elastic ring is shown in Figure 2.

When there are vibrates and deforms of the HRG, the normal displacement and tangential displacement of mass point P are w and V, respectively. In Reference [8], the motion equation of hemispherical resonator in inertial workspace is derived as follows:(1)$${\ddot{w}}^{\prime \prime }-{\ddot{w}}^{\prime \prime }+4\Omega {\dot{w}}^{\prime }+\alpha^{2}\left(w^{\left(6\right)}+2w^{\left(4\right)}+w^{\prime \prime }\right)+\alpha^{2}\xi \left({\dot{w}}^{\left(6\right)}+2{\dot{w}}^{\left(4\right)}+\dot{w^{\prime \prime }}\right)=\left({p^{\prime \prime }}_{w}-{p^{\prime }}_{v}\right)/(\rho S)$$
w(φ,t) is the normal displacement of the ring. Ω is the angular rate of the ring, and its direction is perpendicular to the ring. $\alpha^{2}=EI/(\rho SR^{4})$, *E* is Young’s modulus, *I* is moment of inertia, ρ is material density, *S* is the cross sectional area of ring, *R* is the ring radius. $p_{w}$ and $p_{v}$ are the projection of load on the normal and the tangent of ring. ξ is the damping ratio coefficient of free vibration.

The ring is a straight structure, so the mathematical derivation and model are relatively simple, and it is easy to lead various error into the dynamic model. The ring model of the hemispherical resonator can reflect the basic properties of standing waves, and can describe the motion state of the hemispherical resonator within a certain range of use. However, there are also disadvantages: the precession proportional coefficient K of the hemispherical resonator is seriously distorted, and there is an uncertain moment of inertia I for the hemispherical resonator in the model, which makes it impossible to calculate the movement of the hemispherical resonator beyond the boundary of the edge plane.

In the inertial workspace, the hemispherical resonator rotates at angular velocity. To analyze the vibration law and vibration equation of the hemispherical resonator, the equation of hemispherical resonator is established, the solution of differential equation must be solved by the Bubonov-Galyolkin method. The vibration diagram of the ring model is shown in Figure 3, X1,Y1 are Cartesian coordinate systems, and X2,Y2 are surface coordinate systems.

In the differential equation of ring, the differential equation of the ring without the rotation of the ring model, the external force and the attenuation factor is:(2)$${\ddot{w}}^{\prime \prime }-\ddot{w}+4\Omega {\dot{w}}^{\prime }+\alpha^{2}\left(w^{\left(6\right)}+2w^{\left(4\right)}+w^{\prime \prime }\right)=0$$

From the Reference [9], the optimum vibration mode of HRG is the second-order mode, the solution of differential equation is:(3)w(φ,t)=x(t)cos2φ+y(t)sin2φ
(4)$$\begin{array}{l} \varepsilon (\varphi ,t)={\ddot{w}}^{\prime \prime }-\ddot{w}+4\Omega {\dot{w}}^{\prime }+\alpha^{2}\left(w^{\left(6\right)}+2w^{\left(4\right)}+w^{\prime \prime }\right)\\ =-5\ddot{x}(t)\cdot \cos2\varphi -5\ddot{y}(t)\cdot \sin2\varphi -8\Omega \dot{x}(t)\cdot \sin2\varphi +8\Omega \dot{y}(t)\cdot \cos2\varphi -36\alpha^{2}(x(t)\cdot \cos2\varphi +y(t)\cdot \sin2\varphi ) \end{array}$$

In order to minimize the error of function, it is required to:(5)$$\int_{0}^{2\pi }\varepsilon (\varphi ,t)\cos2\varphi d\varphi =0,\int_{0}^{2\pi }\varepsilon (\varphi ,t)\sin2\varphi d\varphi =0$$
which can obtain:(6)$$\begin{array}{l} \int_{0}^{2\pi }\varepsilon (\varphi ,t)\cos2\varphi d\varphi =\int_{0}^{2\pi }[-5\ddot{x}(t)\cdot \cos^{2}2\varphi -\frac{5}{2}\ddot{y}(t)\cdot \sin4\varphi -4\Omega \dot{x}(t)\cdot \sin4\varphi \\ +8\Omega \dot{y}(t)\cdot \cos^{2}2\varphi -36\alpha^{2}(x(t)\cdot \cos^{2}2\varphi +\frac{1}{2}y(t)\cdot \sin4\varphi )]d\varphi \\ =-5\ddot{x}(t)\cdot \pi +\frac{5}{2}\ddot{y}(t)\cdot 0-4\Omega \dot{x}(t)\cdot 0+8\Omega \dot{y}(t)\cdot \pi -36\alpha^{2}(x(t)\cdot \pi +\frac{1}{2}y(t)\cdot 0)\\ =-5\pi \ddot{x}(t)+8\pi \Omega \dot{y}(t)-36\pi \alpha^{2}x(t)=0 \end{array}$$

It can be concluded that:(7)$$-5\ddot{x}(t)+8\Omega \dot{y}(t)-36\alpha^{2}x(t)=0$$

Similarly:(8)$$5\ddot{y}(t)+8\Omega \dot{x}(t)+36\alpha^{2}x(t)=0$$

Therefore, the dynamic equation of the second-order natural mode of the ideal hemispheric resonator is [10]:(9)$$\{\begin{array}{c} \ddot{x}(t)-\frac{8}{5}\Omega \dot{y}(t)+\frac{36}{5}\alpha^{2}x(t)=0\\ \ddot{y}(t)+\frac{8}{5}\Omega \dot{x}(t)+\frac{36}{5}\alpha^{2}x(t)=0 \end{array}$$

### 2.3. Frequency Splitting Caused by Uneven Density

In order to correct the mass of hemispherical resonator, it is necessary to accurately identify the uneven mass of hemispherical resonator [11]. The uneven mass of the hemispheric resonator can be described by Fourier expansion [12,13]:(10)$$\begin{array}{ll} m & =m_{0}(1+\sum_{n}m_{n}\cos n(\varphi -\theta_{n}))\\ & =m_{0}(1+\sum_{n}m_{n}(\cos n\varphi \cos n\theta_{n}+\sin n\varphi \sin n\theta_{n})) \end{array}$$
where m is the mass of hemispherical resonator. $m_{0}$ is the ideal mass of hemispherical resonator. $m_{n}$ is the harmonic factor of Fourier expansion with uneven mass. φ is the angle of circumference of the hemispheric resonator. $\theta_{n}$ is the azimuth of harmonic mass.

If the structure of hemispherical resonator is the ideal hemispherical shape, the uneven density of hemispherical resonator will cause the uneven mass. The density unevenness of the hemispherical resonator is Fourier expansion along the azimuth, it is
(11)$$\begin{array}{ll} \rho (\varphi ) & =\rho_{0}(1+\sum_{n}\rho_{n}\cos n(\varphi -\varphi_{n}))\\ & =\rho_{0}(1+\sum_{n}\rho_{n}(\cos n\varphi \cos n\varphi_{n}+\sin n\varphi \sin n\varphi_{n})) \end{array}$$
where $\rho_{0}$ is the average density of the material. $\rho_{n}$ is the *n*-th harmonic amplitude of density. $\varphi_{n}$ is the phase of the *n*-th harmonic component of density, n=1,2,3,4…

The following equation can be obtained which ignore high-order term error and carry out series expansion:(12)$$\begin{array}{l} {\ddot{w}}^{\prime \prime }-\ddot{w}+4\Omega {\dot{w}}^{\prime }+\alpha^{2}\left(w^{\left(6\right)}+2w^{\left(4\right)}+w^{\prime \prime }\right)+\alpha^{2}\xi \left({\dot{w}}^{\left(6\right)}+2{\dot{w}}^{\left(4\right)}+\dot{w^{\prime \prime }}\right)\\ ={\ddot{w}}^{\prime \prime }-\ddot{w}+4\Omega {\dot{w}}^{\prime }+\frac{EI}{\rho SR^{4}}\left(w^{\left(6\right)}+2w^{\left(4\right)}+w^{\prime \prime }\right)+\frac{EI}{\rho SR^{4}}\xi \left({\dot{w}}^{\left(6\right)}+2{\dot{w}}^{\left(4\right)}+\dot{w^{\prime \prime }}\right)\\ ={\ddot{w}}^{\prime \prime }-\ddot{w}+4\Omega {\dot{w}}^{\prime }+\frac{1}{\left(1+\sum_{n}\rho_{n}(\cos n\varphi \cos n\varphi_{n}+\sin n\varphi \sin n\varphi_{n})\right)}\alpha^{2}\left(w^{\left(6\right)}+2w^{\left(4\right)}+w^{\prime \prime }\right)\\ +\frac{1}{\left(1+\sum_{n}\rho_{n}(\cos n\varphi \cos n\varphi_{n}+\sin n\varphi \sin n\varphi_{n})\right)}\alpha^{2}\xi \left({\dot{w}}^{\left(6\right)}+2{\dot{w}}^{\left(4\right)}+\dot{w^{\prime \prime }}\right)\\ ={\ddot{w}}^{\prime \prime }-\ddot{w}+4\Omega {\dot{w}}^{\prime }+(1-\sum_{n}\rho_{n}(\cos n\varphi \cos n\varphi_{n}+\sin n\varphi \sin n\varphi_{n}))\alpha^{2}\left(w^{\left(6\right)}+2w^{\left(4\right)}+w^{\prime \prime }\right)\\ +(1-\sum_{n}\rho_{n}(\cos n\varphi \cos n\varphi_{n}+\sin n\varphi \sin n\varphi_{n}))\alpha^{2}\xi \left({\dot{w}}^{\left(6\right)}+2{\dot{w}}^{\left(4\right)}+\dot{w^{\prime \prime }}\right) \end{array}$$

The performance and Q of hemispheric resonant under vibration condition are due to the 3rd harmonic of hemispheric resonant, and the frequency splitting of HRG caused by the uneven density of hemispherical resonator is mainly caused by the fourth harmonic of the uneven density, so the fourth component of the density Fourier expansion is retained [14].

According to the principle of error minimization, the dynamic equation of the second-order natural mode of hemispherical resonator with uneven density distribution is:(13)$$\{\begin{array}{c} \ddot{x}(t)-(\frac{8}{5}\Omega -\frac{18}{5}\alpha^{2}\xi \rho_{4}\sin4\varphi_{4})\dot{y}(t)+\frac{36}{5}\alpha^{2}(1-\frac{1}{2}\rho_{4}\cos4\varphi_{4})x(t)=0\\ \ddot{y}(t)+(\frac{8}{5}\Omega -\frac{18}{5}\alpha^{2}\xi \rho_{4}\sin4\varphi_{4})\dot{x}(t)+\frac{36}{5}\alpha^{2}(1+\frac{1}{2}\rho_{4}\cos4\varphi_{4})y(t)=0 \end{array}$$
where $f_{1}^{2}=\frac{36}{5}\alpha^{2}(1-\rho_{4}\cos4\varphi_{4}),f_{2}^{2}=\frac{36}{5}\alpha^{2}(1+\rho_{4}\cos4\varphi_{4})$.

Thus, the frequency splitting of HRG caused by uneven density can be obtained:(14)$$\begin{array}{ll} \Delta f & =\left|\sqrt{\frac{36}{5}\alpha^{2}(1-\rho_{4}\cos4\varphi_{4})}-\sqrt{\frac{36}{5}\alpha^{2}(1+\rho_{4}\cos4\varphi_{4})}\right|\\ & \approx \sqrt{\frac{36}{5}\alpha^{2}}\cdot \frac{1}{2}\rho_{4}\cos4\varphi_{4}=\frac{1}{2}f_{0}\rho_{4}\cos4\varphi_{4} \end{array}$$
where $f_{0}=\frac{6}{\sqrt{5}}\alpha$ is the natural frequency of hemispherical resonator [15].

### 2.4. The Modal Analysis of Hemispherical Resonator with Uneven Mass

In the differential equation of ring, without the rotation of the ring model, the external force and the attenuation factor, the differential equation of the ring is:(15)$${\ddot{w}}^{\prime \prime }-\ddot{w}+4\Omega {\dot{w}}^{\prime }+\alpha^{2}\left(w^{\left(6\right)}+2w^{\left(4\right)}+w^{\prime \prime }\right)=0$$

According to the reference [16], the optimal circumferential wave number is n=2, if the resonant frequencies of the two modes are inconsistent at this time, frequency splitting will occur. Therefore, it is necessary to analyze the mode of hemispherical resonator to observe the frequency splitting caused by uneven mass.

#### 2.4.1. The Mode of Perfect Hemispherical Resonator

At present, the high precision of HRG usually uses quartz glass as the substrate [17], Young’s modulus is E=76,700 MPa, the density is $2.2\times 10^{3}({\mathrm{kg}/m}^{3})$, and Poisson’s ratio is 0.17, which is shown in Table 1. The perfect hemispherical resonator is used by ANSYS15.0 (64 bit, John, Swanson, PA, USA) software to draw the two-dimensional of hemispherical resonator, and then rotates to obtain the three-dimensional of hemispherical resonator model, the diameter of hemispherical resonator is 30 mm, and the wall thickness is 1 mm [18]. The three-dimensional model of hemispherical resonator is shown in Figure 4.

In the paper, the bottom is fixed, and the top is free to simulate the actual constraints of hemispherical resonator. The modal analysis method of hemispherical resonator is carried out in ANSYS to obtain the mode of the hemispherical resonator and its corresponding natural frequency as shown in Figure 5.

In the vibration mode of HRG, the zero-order mode is the tension-compression vibration of hemispherical resonator, the first-order mode is the swing of hemispherical resonator in a fixed direction, and the second-order mode is the mode of vibration with a phase angle difference of 45°. It can be seen from the analysis of figure that mode 4 and mode 5 of the hemispheric resonator correspond to the second-order mode [19], the natural frequencies are, respectively, 5663.5 Hz and 5663.6 Hz, and the frequency splitting is 0.1018 Hz.

#### 2.4.2. The Mode of Hemispherical Resonator with the Uneven Mass

The uneven mass is mainly reflected in the uneven mass of circumferential caused by the local position uneven mass of hemispherical resonator. The modal analysis of uneven mass is realized by simulating the uneven mass of circumferential in ANAYS [20]. Four asymmetric positions are selected on the lip edge of the three-dimensional model of hemispherical resonator, and a rectangular block with a length of 1 mm, a width of 1 mm and a height of 0.5 mm is added to obtain the uneven mass of hemispherical resonator [21]. Additionally, the equivalent mass value of each block added is $0.001125\times 10^{-3}$ kg. The three-dimensional solid model of hemispherical resonator is shown in Figure 6:

Similarly, the fixed position constraint is added to the hemispherical resonator, and the freedom of bottom is limited in ANSYS. Then, the modal analysis is conducted to obtain the natural frequency of the second-order mode of the uneven mass. It is show in Figure 7.

It can be seen from Figure 7 that the natural frequencies of mode 4 and mode 5 are 5640.1 Hz and 5659.2 Hz, respectively, and the frequency splitting is 19.1414 Hz. It shows that the uneven mass of hemispherical resonator causes the obvious frequency splitting.

## 3. The Identification of Harmonic Component with the Uneven Mass

The machining accuracy and quality of hemispherical resonator directly restrict the working performance of HRG. There are some problems in its machining, such as long cycle, low efficiency, high cost, uneven mass and excessive residual stress [22]. Airbag, magneto rheological and plasma can be used for the machining technologies of hemispherical resonator, but there are some defects in every processing method, which are the result of uneven mass of hemispherical resonator.

### 3.1. The Identification of 4th Harmonic of Mass

The error of uneven mass is mainly caused by the process defects of the hemispherical resonator. The first three harmonics of the hemispherical resonator with the uneven mass will affect the quality factor, increase the sensitivity of the hemispherical resonator to external vibration, increase the error of random walk and drift, and reduce the anti-disturbance and accuracy of HRG. However, the 4th harmonic of Fourier expansion of the parameters such as the circumferential density, Young’s modulus and the wall thickness of the hemispherical resonator has a great impact on the standing wave characteristics. If the mass distribution of hemispherical resonator is uneven, there will be 4th-order mass defects and orthogonal errors. In order to compensate the drift of HRG, the hemispherical resonator must be balanced according to the 4th harmonic of the deviation to reduce the splitting value of the natural frequency [23].

The mass balance can be achieved by removing the excess mass of hemispherical resonator. When the change of shape and position error of hemispherical resonator is not considered, the frequency splitting of hemispherical resonator can be controlled by identifying the 4th harmonic component of its relative density.

At present, the processing of hemispheric resonator adopts molten quartz glass, the processing error of shape and position is in micron level, and the natural frequency is between 4000~5000 Hz. In the paper, the natural frequency of hemispheric resonator is 4800 Hz, the 4th harmonic amplitude is 1 μm. When the frequency splitting is $\Delta f=\frac{1}{2}f_{0}\rho_{4}\cos4\varphi_{4}$, the identification of amplitude and phase of 4th harmonic is needed.

The dynamic equation of second-order natural mode of hemispheric resonator with uneven density distribution:(16)$$\{\begin{array}{c} \frac{18}{5}\alpha^{2}\xi \rho_{4}\sin4\varphi_{4}\dot{y}(t)-\frac{18}{5}\alpha^{2}\rho_{4}\cos4\varphi_{4}x(t)=-\ddot{x}(t)+\frac{8}{5}\Omega \dot{y}(t)-\frac{36}{5}\alpha^{2}x(t)\\ -\frac{18}{5}\alpha^{2}\xi \rho_{4}\sin4\varphi_{4}\dot{x}(t)+\frac{18}{5}\alpha^{2}\rho_{4}\cos4\varphi_{4}y(t)=-\ddot{y}(t)-\frac{8}{5}\Omega \dot{x}(t)-\frac{36}{5}\alpha^{2}y(t) \end{array}$$

The variables are:$$X=\rho_{4}\sin4\varphi_{4},Y=\rho_{4}\cos4\varphi_{4}x(t)\;a_{11}=\frac{18}{5}\alpha^{2}\xi \dot{y}(t),a_{12}=\frac{18}{5}\alpha^{2}x(t),a_{21}=-\frac{18}{5}\alpha^{2}\xi \dot{x}(t),a_{22}=\frac{18}{5}\alpha^{2}y(t)\;b_{1}=-\ddot{x}(t)+\frac{8}{5}\Omega \dot{y}(t)-\frac{36}{5}\alpha^{2}x(t),b_{2}=-\ddot{y}(t)-\frac{8}{5}\Omega \dot{x}(t)-\frac{36}{5}\alpha^{2}y(t)$$

Then, the dynamics equation is changed into a simple linear equation:(17)$$\left\{ \begin{array}{l} a_{11}X+a_{12}Y=b_{1}\\ a_{21}X+a_{22}Y=b_{2} \end{array}\right.$$

After solving the equation, it can get:$$\rho_{4}=\sqrt{X^{2}+Y^{2}},\varphi_{4}=\frac{1}{4}\arctan(\frac{Y}{X})$$

### 3.2. The Adjustment of Frequency by the Mass Division

The uneven mass causes the frequency splitting of hemispheric resonator, and the frequency splitting can also be optimized by changing the mass distribution, that is, the frequency can be adjusted by the mass [24,25].

At present, the manufacturing process of hemispherical resonator, especially the ultra-precision machining process and leveling process of hemispherical resonator, are rarely reported. In recent years, the manufacture method of hemispherical resonator mostly adopts the ion beam method, that is, the sputtering effect is used to remove the excess material at the local position of hemispherical resonator. The quality of ion beam leveling is basically linear with the leveling time, and the removal mass can be controlled in micrograms [26,27,28,29].

After machining and chemical processing, the hemispherical resonator generally has certain mass poles. It is assumed that the characterization value of mass balance parameter is $m_{kc}$ and $m_{ks}$, the initial angle of partition region is $\varphi_{i}^{L}$ and final angle is $\varphi_{i}^{R}$, where the defect number is k=1,2,3…k and the clustering mesh division number is i=1,2,3…N.

The balance caused by uneven mass is:(18)$$m_{kc}=\int_{0}^{2\pi }\Delta m(\varphi )\cos k\varphi d\varphi ,\;m_{ks}=\int_{0}^{2\pi }\Delta m(\varphi )\sin k\varphi d\varphi$$

The mass distribution function of the same layer of material removed is:(19)$$\Delta m(\varphi )=\left\{ \begin{array}{r} \rho ,\varphi \in \left[\varphi_{i}^{R};\varphi_{i}^{L}\right]\\ 0,\varphi \notin \left[\varphi_{i}^{R};\varphi_{i}^{L}\right] \end{array}\right.$$

$\varphi_{i}^{L}$ and $\varphi_{i}^{R}$ are the sector geometry angles, and $m_{kc}^{*}$ and $m_{ks}^{*}$ are the actual mass balance parameter, so the optimal mass is:(20)$$M_{\mathrm{optimal}\;}=\min(\frac{1}{2}\sum_{k=1}^{K}\left[{\left(m_{kc}-m_{kc}^{*}\right)}^{2}+{\left(m_{ks}-m_{ks}^{*}\right)}^{2}\right])$$

Calculate the derivative $M(\varphi_{i}^{R},\varphi_{i}^{L})$ as:(21)$$\frac{\partial M_{\mathrm{optimal}\;}}{\partial \varphi_{i}^{R}}=\sum_{k=1}^{K}\left[(m_{kc}-m_{kc}^{*})\cos k\varphi_{i}^{R}\right]-\sum_{k=1}^{K}\left[(m_{ks}-m_{ks}^{*})\sin k\varphi_{i}^{R}\right]$$
(22)$$\frac{\partial M_{\mathrm{optimal}\;}}{\partial \varphi_{i}^{L}}=\sum_{k=1}^{K}\left[(m_{kc}-m_{kc}^{*})\cos k\varphi_{i}^{L}\right]+\sum_{k=1}^{K}\left[(m_{ks}-m_{ks}^{*})\sin k\varphi_{i}^{L}\right]$$

Then, the mass to be removed can be obtained:(23)$$dM_{\mathrm{optimal}}=\sum_{i=1}^{K}\frac{\partial M_{\mathrm{optimal}}}{\partial \varphi_{i}^{R}}d\varphi_{i}^{R}+\sum_{i=1}^{N}\frac{\partial M_{\mathrm{optimal}}}{\partial \varphi_{i}^{L}}d\varphi_{i}^{L}$$

Iterative calculations are used to determine the removed unbalanced mass and to ensure that $M_{\mathrm{optimal}}+dM_{\mathrm{optimal}}=0$, until $M_{\mathrm{optimal}}<\varepsilon$ completes the first cycle.

The cycle is finally removed mass is:(24)$$M=\int_{0}^{2\pi }\Delta m(\varphi )d\varphi =\sum_{i=1}^{N}\int_{\varphi_{i}^{R}}^{\varphi_{i}^{L}}\rho d\varphi =\rho \sum_{i=1}^{N}\left(\varphi_{i}^{L}-\varphi_{i}^{R}\right)$$

The mass of hemispherical resonator is not completely balanced after the first cycle, and the accuracy of each leveling process has an impact on the next one, so the accuracy of the hemispherical resonator needs to be achieved through multiple cycles.

### 3.3. The Simulation of Mass Balance by ANASY

The 4th harmonic mass is the main source of Fourier error of unbalanced mass when HRG operating in *n* = 2 mode. Each harmonic distribution has a mass concentration region with the same order as its Fourier expansion. The distribution of the 1st~4th harmonic mass along the circumferential direction of hemispherical resonator is shown in Figure 8. The manifestations of uneven mass are that the first harmonic is an eccentric circle, the second harmonic is an ellipse, the third harmonic is a three-sided circle, and the fourth harmonic is a four-sided circle [30].

The processing of high precision hemispherical resonator is to adopt one-time forming process on fused quartz glass materials with high hardness and high brittleness, and to realize the “three centers in one” and axial symmetry of mass center. The center of gravity and shape of hemispherical resonator must have repeated grinding to improve the processing accuracy. In order to minimize the influence of uneven mass on the measurement precision of HRG, it is necessary to realize that the hemispherical resonator has high-precision uniformity. The frequency splitting caused by the 4th harmonic is one order of magnitude larger than the other harmonics [31]. Therefore, it should avoid the quadrangular circle caused by the 4th harmonic.

At present, the mass distribution of hemispherical resonator will be uneven due to subsurface defects on the surface of hemispherical resonator. In order to simplify the simulation model, a bulge is added at four asymmetric positions along the lip of the hemispherical resonator to simulate the state of uneven mass, which is set as a rectangular block with length of 1 mm, width of 1 mm and height of 0.5 mm. By simulating the mass leveling process, a layer of excess material with a thickness of 0.05 mm is evenly removed from the blocks to compensate the non-uniform resonance of the hemispherical shell and improve the non-equilibrium parameters of the hemispherical resonator [32,33,34]. Observe the change of frequency splitting, and the change of mass leveling frequency splitting is shown in Table 2.

It can be seen from the table that the removal height of mass increases, the frequency splitting decreases until the thickness reaches 0.50 mm, that is, when all the convex parts are removed, the frequency splitting is consistent with the frequency splitting of the perfect of hemispherical resonator, so frequency splitting was effectively inhibited.

## 4. Conclusions

In the paper, the frequency splitting caused by the uneven mass of hemispherical resonator in the machining process of hemispherical resonator and other factors is analyzed. The model of hemispherical resonator is established by using ANSYS software. Additionally, the mode of hemispherical resonator in ideal state and uneven mass are simulated, and the influence of 4th harmonic of density on frequency splitting is analyzed. By establishing the dynamic equation of the second-order natural mode of hemispherical resonator with uneven density distribution, the frequency splitting caused by uneven density distribution is obtained, and the 4th harmonic of density is identified. The method of partial mass removal is used to suppress the frequency splitting of HRG, and the simulation results show that the method is effective. Therefore, in the actual machining process of hemispherical resonator, the mass balance should be realized as much as possible to avoid various surface damage, internal defects and uneven mass distribution in the machining process.

## Figures and Tables

**Figure 1 micromachines-13-02015-f001:**
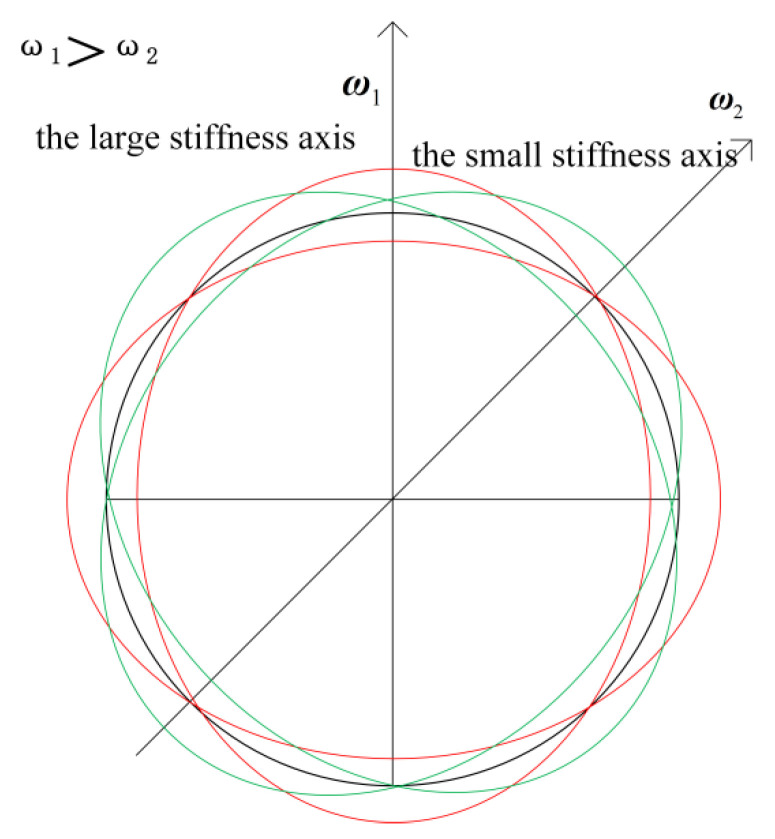
The stiffness axis of m-HRG.

**Figure 2 micromachines-13-02015-f002:**
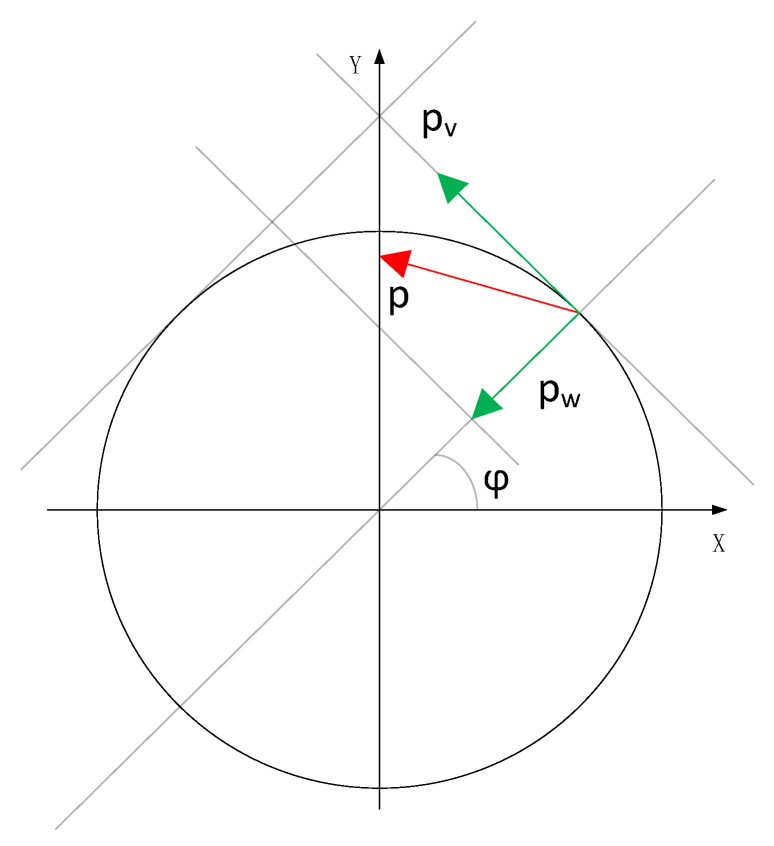
The model of elastic ring.

**Figure 3 micromachines-13-02015-f003:**
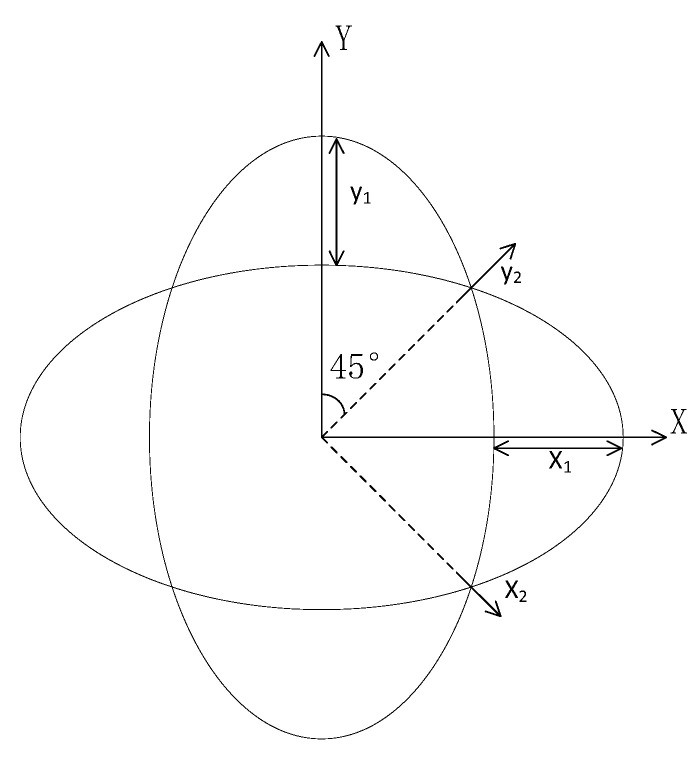
The vibration diagram of the ring model.

**Figure 4 micromachines-13-02015-f004:**
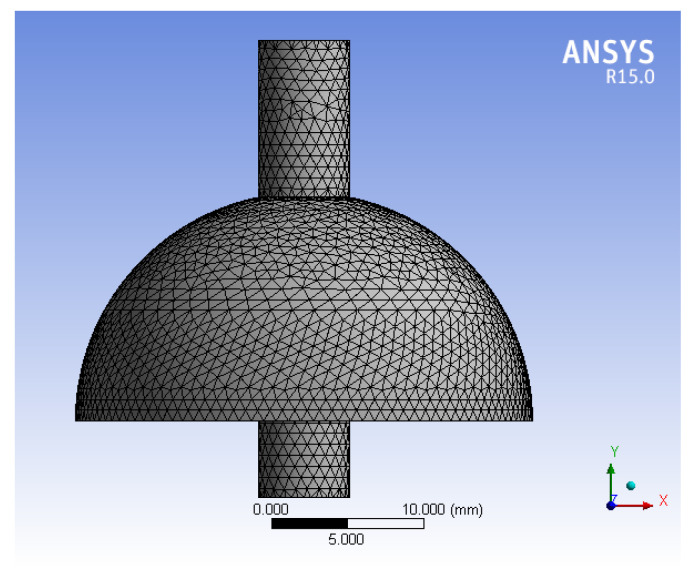
The three-dimensional model of hemispherical resonator.

**Figure 5 micromachines-13-02015-f005:**
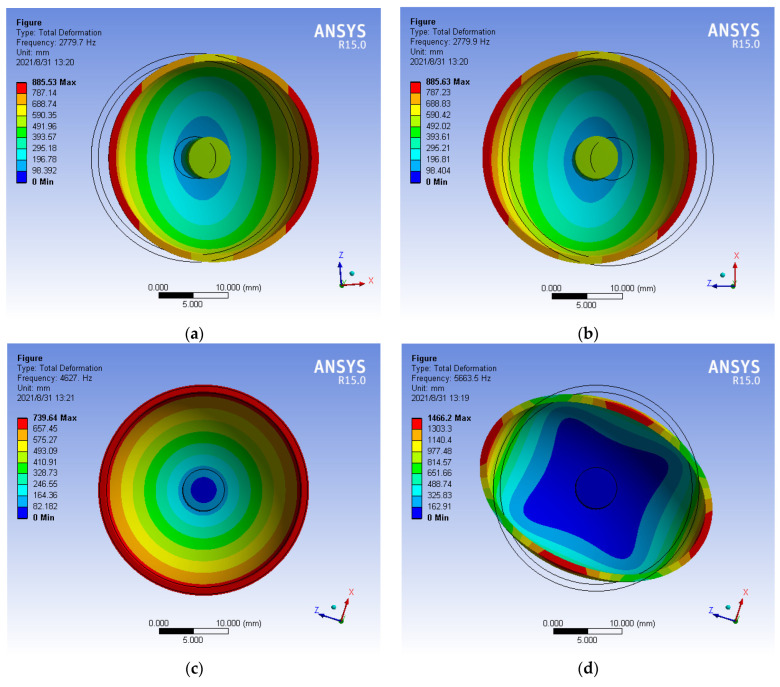
(**a**) Mode 1 (first order, natural frequency 2779.7 Hz); (**b**) Mode 2 (first order, natural frequency 2779.9 Hz); (**c**) Mode 3 (zero order, natural frequency 4626.9 Hz); (**d**) Mode 4 (second order, natural frequency 5663.5 Hz); (**e**) Mode 5 (second order, natural frequency 5663.6 Hz).

**Figure 6 micromachines-13-02015-f006:**
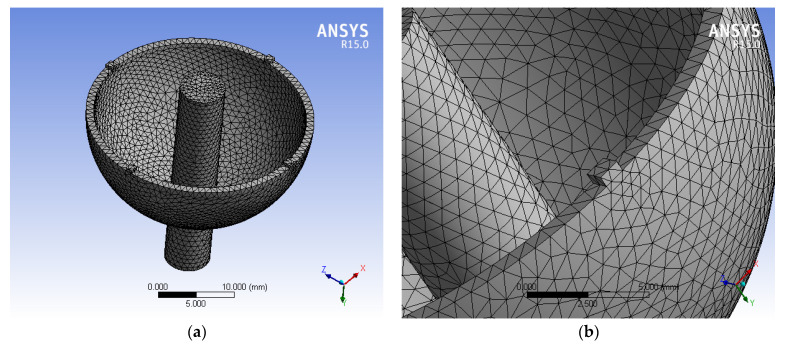
The uneven mass of hemispherical resonator. (**a**) The concentrated mass Solid model of hemispherical resonator; (**b**) Lumped mass of hemispherical resonator.

**Figure 7 micromachines-13-02015-f007:**
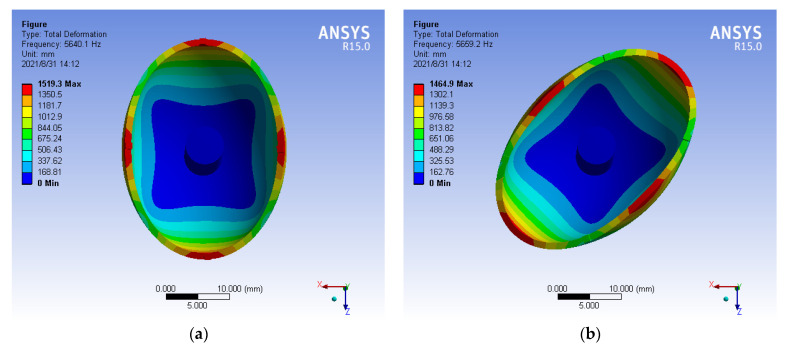
(**a**) Mode 4 (natural frequency 5640.1); (**b**) Mode 5 (natural frequency 5659.2 Hz).

**Figure 8 micromachines-13-02015-f008:**
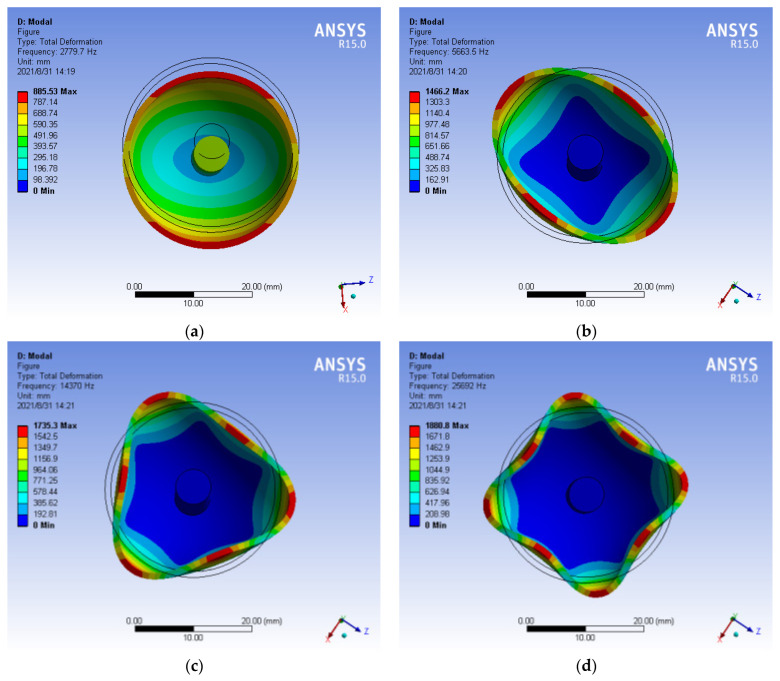
The manifestation of each harmonic with uneven mass distribution. (**a**) First harmonic; (**b**) Second harmonic; (**c**) Third harmonic; (**d**) Fourth harmonic.

**Table 1 micromachines-13-02015-t001:** Dimension paramenters.

Structural Parameters	Diameter of Harmonic Oscillator (mm)	Shell Thickness (mm)	Young’s Modulus (MPa)	Density (kg/m^3^)	Poisson’s Ratio
Parameter value	30	1	766,700	2200	0.17

**Table 2 micromachines-13-02015-t002:** The removal height of mass and its corresponding frequency splitting.

Steps	The Removal Height of Mass (mm)	The Natural Frequency of Mode 4 (Hz)	The Natural Frequency of Mode 5 (Hz)	Frequency Splitting Δf(Hz)
0	0	5640.104155855	5659.245586609	19.141430754
1	0.05	5642.839856892	5659.742846148	16.902989256
2	0.10	5645.646469793	5660.507782266	14.861312473
3	0.15	5647.897040007	5660.966849323	13.069809316
4	0.20	5650.947992024	5661.454239987	10.506247963
5	0.25	5652.929648874	5661.166872808	8.237223934
6	0.30	5655.672389432	5661.661513544	5.989124112
7	0.35	5658.416513497	5662.039588951	3.623075454
8	0.40	5659.866145712	5661.307069927	1.440924215
9	0.45	5661.663708925	5662.006946328	0.343237403
10	0.50	5663.466956867	5663.568805673	0.101848806

## Data Availability

Not applicable.

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
