# Peer review of "Research of Frequency Splitting Caused by Uneven Mass of Micro-Hemispherical Resonator Gyro"

_micromachines, 2022, doi:10.3390/mi13112015_

Round 1

Reviewer 1 Report

The Authors propose an analysis on the effects of an uneven mass distribution on the resonant frequency of the micro–Hemispherical Resonant Gyroscope. Although the results are promising for the next generation of gyros, a major revision of the manuscript is suggested to address the following comments:

-          In the Introduction Section, the Authors should discuss about the potential applications and their target performance to help the reader in the device rating. Moreover, the discussion on the competing technologies should be improved. As example, the Authors should highlight the benefits of different resonant gyros, whose key element is a high Q factor ring resonator (e.g. “Comprehensive mathematical modelling of ultra-high Q grating-assisted ring resonators,” Journal of Optics, 22(3), 035802, 2020; “Integrated waveguide coupled Si3N4 resonators in the ultrahigh-Q regime,” Optica, 1(3), 153-157, 2014; “422 Million intrinsic quality factor planar integrated all-waveguide resonator with sub-MHz linewidth,” Nature communications, 12(1), 1-8, 2021) also comparing the achieved results with the state-of-the-art. This could be useful to understand the benefit of choosing this kind of gyros among the others.

-          In the introduction section it has been said that the frequency splitting damage the performances of the gyro but it should be explained in what way and how it is relevance in order to underline the importance of the mass distribution.

-          In the paragraph 2.2 the vibration of m-HRG is modelled through thin elastic ring vibrations. The Authors should explain the benefits of this simplification and which parameter has been neglected.

Reviewer 2 Report

The Hemispherical Resonant Gyroscope (HRG) has been developing in recent years, attracting significant interest and is deemed as a promising technique for the evolution of mico-resonant gyroscope. This paper reports the analytical evaluation of uneven mass which is caused by the fabrication tolerances and other factors, providing a comprehensive theoretical derivation in terms of vibration modes and behaviours. The resultant asymmetrical mass distribution (or density) will create inevitable errors and instability of the HRG in practical application, impacting the performance of such a device. A possible mass removal method is also proposed to compensate for the uneven mass, alleviating the frequency splitting. In this context, high-precision HRG is of considerable potential to be achieved, if the mass distribution can be trimmed accurately and effectively. Thanks for submitting this interesting work, it offers a prospective method to determine the consequences of uneven mass of the HRG, as well as a correction approach with respect to the simulations. However, there are some comments that have to be addressed.

Comments:

1.      Concerning the whole text, the English writing is a little brusque, for example, in line 17, “ANSYS software which can accurately reflect”, here replacing reflect to mimic is better. In line 32, “the main component is high-precision hemispherical resonator” should be “the main component of which is a high-precision hemispherical resonator”. In line 62, “and there is no problem of frequency 62 splitting, that is, the frequency splitting is 0”, to “therefore the problem of frequency splitting is averted.”

2.      In line 37, it is better to give author names, rather than reference numbers, same for line 39.

3.      In lines 50 and 53, the sentences are repeated.

4.      Figure 1 is kind of vague, authors can use two figures, one with large stiffness and another one with small stiffness. Also, please label the curves and axis, as well using solid and dashed lines to discern the different vibration modes.

5.      In line 78, “which is made of fused quartz glass with high precision”, please give a brief introduction of the fabrication tolerances, or put a reference here to define the high precision of the manufacturing.

6.      In figure 2, the authors can define the direction of axes as well as indicate the centre of the ring, assisting the reader to understand the ring model.

7.      In line 92, damping time is correct? Or it is a damping ratio coefficient?

8.      In line 96, “must be solved by the Bubonov-Galyolkin”, should be “must be solved by the Bubonov-Galyolkin method”.

9.      In figure 3, please explain y1,2 and x1,2, since they do not appear in the subsequent sections.

10.   In line 143, there could be a typing error or display error of the equation.

11.   In line 152, seems the equation is absent.

12.   In section 2.4.1, a table that lists the dimensional parameters of the resonator, and the supporting rod, can help the reader to understand the simulation.

13.   In line 193, rectangular blocks are added, it is better to calculate the equivalent mass value of each block.

14.   In section 2.4.2, the simulation is based on an asymmetric position with the same extra mass, , in turn, what will happen if the position is symmetrical but the mass values are different?

15.   In lines 198 and 199, the caption of figure 6, please rephrase the “solid model”, which is vague. Also, the lumped mass.

16.   In lines 219 and 220, it is better to give a reference here so that the reader can have a source to know why and how the harmonics affect the HRG.

17.   In line 284, the “accuracy” is vague, this accuracy is related to what? Fabrication? Performance?

18.   In line 307, please give a reference or a simulation result, verifying the frequency splitting of the 4th harmonic is one order of magnitude larger than the other harmonics.

19.   In lines 327 and 328, “caused by” is repeated twice.

20.   Regarding the technique contents, Table 1 is more correlated with section 2.4.2, the authors introduce a removal of a layer of mass to compensate for the frequency splitting. And the simulation results concern mode 4 and mode 5 only.

21.   In addition, the structure of the submitted manuscript can be refined, for example, authors can introduce and explain the mass removal method in section 2.4.2, as well as demonstrate the simulation results. Then in the following sections, explain the influences of the higher order of harmonics.

22.   This also leads to another question, as a reader, I do not quite understand the contents regarding the 4th harmonic, the authors use exactly the same simulation model as in section 2.4.2, with blocks of additional mass. In practice, the 4th harmonic of the HRG should be suppressed or avoided? If so, a proper method or possible approaches should be included.

23.   Please retain uniformity of the reference format, for instance, some are with full names, some with abbreviations, and some are ignored.

Round 2

Reviewer 1 Report

The Authors have modified the manuscript according to the Reviewer suggestions. However, the comparison with respect the most performing gyros, as RMOG, RLG should be improved (see in details Comprehensive mathematical modelling of ultra-high Q grating-assisted ring resonators, Journal of Optics, 22(3), 035802, 2020; Integrated waveguide coupled Si3N4 resonators in the ultrahigh-Q regime, Optica, 1(3), 153-157, 2014; 422 Million intrinsic quality factor planar integrated all-waveguide resonator with sub-MHz linewidth, Nature communications, 12(1), 1-8, 2021).

Reviewer 2 Report

The authors have addressed the comments, thanks for the effort to improve the contents of the submitted manuscript.

There is one additional point:

In line 22, “can ensure the high-quality factor”. There is not any specific value given in the paper, and the Q-factor value changes due to uneven mass are not include in the simulation.
